# Retaining Non-EU Immigrants in Rural Areas to Sustain Depopulated Regions: Motives to Remain

Elisete Diogo [1,2]

1  CARE-Research Center on Health and Social Sciences, VALORIZA—Centro de Investigação para a Valorização de Recursos Endógenos, Instituto Politécnico de Portalegre, 7300-110 Portalegre, Portugal; elisetediogo@gmail.com
2  Católica Research Centre for Psychological, Family and Social Wellbeing (CRC-W), Universidade Católica Portuguesa, 1649-023 Lisbon, Portugal

**Abstract:** Rural areas face multiple challenges. Among these are population decline and the attendant economic and social problems, namely demographic issues. Although the factors that draw immigrants to other countries are known, comprehending the factors that result in immigrants remaining in rural areas after their arrival could support informed local policies and practices. The purpose of the study is to explore the motivations that shape immigrants' intentions to stay in Alentejo, a depopulated region in Portugal. The research questions are as follows: What motivates immigrants to remain in depopulated regions in Portugal? Furthermore, what contributions can practitioners and immigrants make to local policies and practices? Practitioners (*n* = 8) and non-European Union immigrants (*n* = 15) living in this region were interviewed between 2020 and 2021. The empirical data were analyzed using the MaxQDA software. The results indicated that the intention to remain in rural areas arises from a progressive process: this is a process that immigrants experience that motivates them to stay there long-term. The factors influencing the process include four components described throughout this work: (1) Instrumental and material motivations; (2) Emotional and social motivations; (3) Motivations based on the quality of life; and (4) Motivations based on the political dimension. The conclusions highlight the implications for policies and practices, suggesting more investment into rural regions to reverse the depopulation trend.

**Keywords:** international immigrants; rural areas; depopulated regions; sustainability; regional development





## 1. Introduction

Rural systems tend to be described as three subsystems: the economic/industry subsystem, the social/population subsystem, and the land/spatial subsystem, which are framed within a social–ecological framework [1]. In this work, we pay attention to several challenges that rural areas face [2]. These systems are characterized by poor access to public services and jobs, low accessibility, a lack of innovation, a lack of economic competitiveness, and precarious governance [3,4]. As a result, in Portugal, a policy of positive discrimination has emerged for so-called "Low-Densty Territories" (LDT). The term is applied to disadvantaged areas based on the populational density, demography, population, socioeconomic and geographic features, and accessibility aspects [5] to support their sustainable development. Sustainable development is increasingly viewed as being the way to promulgate just and practicable economic, environmental, and social policy [6]; therefore, in this article, the contributions of the policymakers are presented.

The objective of the present research is to identify and understand the motivations shaping immigrants' intentions to stay long-term in Alentejo, a depopulated region in Portugal, using qualitative methods. Since most research tends to focus on migration phenomena [7] in large cities and only to a lesser degree in rural areas [8], the present study

is innovative and timely. Moreover, it consists of a contribution to the implementation of the Action Plan on Integration and Inclusion 2021–2027, presented by the European Commission, promoting the inclusion and integration of migrants in rural areas.

*Literature Review*

The issues presented above underlie depopulation caused by a long-standing exodus phenomenon as citizens decide to move to big cities and abroad [7,8]. Both factors that draw people to other cities or countries and those that draw them to leave rural areas contribute to depopulation [9] and to the absence of people and services there [2]. Migration[1] to smaller cities and rural areas can help overcome this depopulation and ensure the viability (or the return) of basic services such as schools, hospitals, and shops, as well as greater diversity, which is associated with economic growth [10–12].

Morén-Alegret and colleagues [11] presented empirical results regarding the demographic, social, and economic challenges faced by rural areas and small villages, arguing that international immigrants add value and play a role in the sustainability and development of rural mountainous areas, namely as immigrant entrepreneurs. Immigrants are transforming these rural challenges into opportunities, as exemplified in the statement, "opening municipal politics up and encouraging respect for others, instead of prejudice, gossip, racism, classism, and sexism" (p. 290).

Although we are studying rural areas and their challenges, such as aging, low population, and so on, the knowledge of each region requires a deep understanding. Thus, it is relevant to understand the socio-demographic and economic characteristics of a concrete rural region.

### Rural Areas—Facts and Figures of Alentejo

The population density in Alentejo region is, on average, 22 inhabitants/km$^2$ [13] (see Table 1). In comparison, the European Commission [13] defines rural areas as areas having a population density of fewer than 300 inhabitants/km$^2$. Portugal has an average of 113 individuals per km$^2$, while the metropolitan area of Lisbon (the capital city) has 952 individuals per km$^2$ [14].

In the region, there was a decrease in the number of inhabitants in 2021 of around 7.0% from a total of 704,533 people [15]. The effect of net migration has not compensated for this natural decrease [8]. An exception is Odemira, a Portuguese municipality located in Alentejo region that has seen the highest population growth in the last 10 years, supported by international immigrants.

From a demographic perspective, the aging index is high in Alentejo, particularly in the interior near the border with Spain, within the rural context [16]. It has recorded an increase in aging (219 elderly people per 100 young people) and a decrease in new population [15]. The birth rate is 7.4, compared to 9.3 in the metropolitan area of Lisbon and 7.7 in the entire country [17].

Observing these economic indicators, Alentejo region has the lowest activity rate (44.6%). There are 291,269 employed individuals and 4,193,900 non-working individuals. [14]. As a result, a few of the municipalities have the highest unemployment rates in the country. The greatest economic activities are commerce, followed by industry, specifically agriculture, the construction industry, the financial industry, and then state activities.

The region has the most derelict housing as well. However, rents tend to be below the national average, at EUR 328 per month. Rents exceeding EUR 1000 are mainly in the metropolitan area of Lisbon [14].

Alentejo stands out not only for having a low crime rate but also for being a safe place to live for both nationals and immigrants. However, human trafficking is noted, as well as labor exploitation, particularly associated with agricultural work [18].

Although immigrants largely live in urban areas, rural areas have increasingly begun to attract immigrants in considerable numbers [9], as in Alentejo, for example. In 2021, there were 23,737 (19.8%) newcomers [19]. In total, there were 52,316 immigrants in Alentejo alone, out of a total of 1,089,023 in the whole country [20,21]. The most represented

nationalities of third-country citizens in Alentejo were Brazilian (10,083), Indian (7383), Nepalese (3659),, Ukrainian (2224), Chinese (1591), British (1434), and Angolan (723) [22].

Although detailed data about immigrants living in Alentejo is not available, national reports [19] state that 77.1% of the immigrants are of working age, especially within the age range of 25–44 years old. Employment is the main reason to request authorization for residence in Portugal. Immigrant children (13.6%) outnumber immigrant old people (9.3%); the opposite is seen within the native population. Men comprise 52.4% of the population.

Initiatives that contribute to halting or reversing the situation and retaining natives attract newcomers may lead to the sustainable development of these regions [9,11,12,23].

**Table 1.** Facts and figures of Alentejo.

|  | *Alentejo region* |
| --- | --- |
| Population density | 22 inhabitants/km$^2$ (113 inhabitants/km$^2$ in Portugal) |
| Inhabitants | Decreasing: 7.0%, Less 704,533 inhabitants |
| Population aging | Increasing: 219 elderly for 100 young people |
| Birth rate | 7.4 (7.7 in Portugal) |
| Economic activities | Commerce, industry, agriculture, construction industry, financial and state activities |
| Economic indicator: active people | 291,269 employees 4,193,900 non-active population |
| Economic activity rate | The lowest of the country: 44.6% |
| Rent | EUR 328 per month on average |
| International immigrants: newcomers in 2021 | 23,737 (19.8%) |
| Non-EU residents | 52,316 (1,089,023 in Portugal) |
| International immigrants by country of origin | Example: Brazil (10,083), India (7383), Nepal (3659), Ukraine (2224), China (1591), the United Kingdom (1434), and Angola (723) |

Source: own elaboration based on [9,13–19,22].

**Why Do Immigrants Remain in Adverse Places?**

Pull factors for immigrants to go to rural and depopulated regions have been highlighted [9]. Nevertheless, understanding the factors for remaining after arriving is central to consolidating a strategic local policy to attract and retain immigrants and an informed practice for social inclusion. A challenge for rural areas is achieving long-term integration, as migrants often seek to leave for bigger cities [10].

Most residents in rural areas are international immigrants who come to work in low-skilled jobs in agriculture, construction, agribusiness, the care of dependent people, and other services [23]. Labor-intensive and export-oriented agriculture acts as a magnet for migrants [9,10,12]. Although employment and job conditions may represent a key motivating factor to stay in rural areas [10,23], decisions are most often based on a combination of material and emotional considerations, balancing experiences and aspirations for the past, present, and future [24].

Flynn and Kay [24] argued that both material (access to jobs and housing), emotional aspects (such as positive relationships), and emotional integration must be addressed. For example, when an immigrant is offered the opportunity of a permanent contract, they may decide to stay and plan to bring family members (e.g., a daughter or an elderly mother) to live with them. Therefore, establishing an important step towards a more permanent stay, including family reunification and bringing in other compatriots, is desirable for sustaining these regions.

Valdez and colleagues [25] presented psychological, social, and political factors that have been used to explain immigrants' transitions in orientation from temporary to permanent settlement in an anti-immigration state in the USA. In terms of personal motivations, they found perseverance, resignation, and fatalism, as well as religious faith, as factors. In

terms of family and community, dreams and hopes for their children, access to community resources, social ties, and belonging resonated. Feeling connected with a community seems to influence immigrants' decisions to settle permanently [26]. Finally, in terms of socioeconomic and political motivations, it has been found that conditions in the host country and the country of origin, including security/ violence issues, as well as perceived sociopolitical constraints in other municipalities, are relevant [25]. Therefore, economic crises, anti-immigration sentiment, and unstable employment opportunities influence the willingness to stay in adverse places.

Rural areas and smaller cities offer advantages, according to Gauci [10], as it is easier to develop a social network that supports immigrant integration. Gauci presents, as an example, the lower risk of school segregation because, for instance, there is only one school. In comparison with bigger cities, there are "greater opportunities for interaction with local communities, a more tight-knit safety net [. . .], integration efforts with lower budgets [. . .], job opportunities in the location, lower property [rental] prices, and reasonable connections [in terms of cost and time]" (p. 31).

Another profile of settled immigrants is relatively high-income individuals/ families [9], namely from the United Kingdom (complementary to other European countries) who, after Brexit, are also considered third-country nationals. More recently, it has been noted that the impact of the changing nature of work, such as the increase in remote working [27], i.e., digital nomads, has also resulted in a greater movement toward people living in smaller municipalities [10]. Quiet and peaceful surroundings are difficult to find in a city, and therefore, a better environment for children is also a motivation to stay in depopulated regions [24].

However, Flynn and Kay [24] noted, "for many of our participants, longer-term stays had not been initially planned, but had gradually emerged, becoming the plan for the foreseeable future" (p. 60). Thus, immigrants may consider alternatives to staying, and then, if the risks and uncertainties are higher in other regions, they remain where they are [25].

## 2. Materials and Methods

The empirical research design [28] is grounded in a qualitative approach supported by interviews. The main research questions [28] include: (1) What motivates an immigrant's decision to remain living in depopulated regions in Portugal? and (2) What contributions can practitioners and immigrants make to local policy and practice?

The theoretical sampling [29,30] was supported by maximum variation and convenience criteria, gathering empirical data until theoretical saturation.

Participants in the study consisted of eight practitioners who worked with non-European Union immigrants/third-country nationals and 15 immigrants who worked and lived in the Alentejo region. Semi-structured interviews were conducted for several months between 2020 and 2021. The length of the interviews was around 1:20 h each, and all were voice-recorded (after permission was given). Conversations were in Portuguese, except one with an immigrant from India, who arrived the year before and had poor Portuguese.

The topics of the interviews were significantly wide, considering the broad research project. Therefore, they consisted of a characterization of the immigration process and families, as well as the integration in the host country. Topics also included practices and activities promoting social inclusion and, finally, suggestions to improve migration policy and to support local policymakers. For this specific study, questions focused particularly on what could set a person in that municipality and what could be attractive for immigrants wanting to stay.

### 2.1. Participants

The group of practitioners (Figure 1) was social worker members of local centers for the support and integration of migrants (named CLAIM) or from municipalities. These

CLAIM and municipality groups operate in the Alentejo region. These practitioners likely have a higher education, i.e., a degree in Social Sciences (Social Work, Social Pedagogy, and Psychology), as well as in Engineering and Architecture. They were predominantly natives of Portugal and comprised 3 males (37.5%) and 5 females (62.5%).

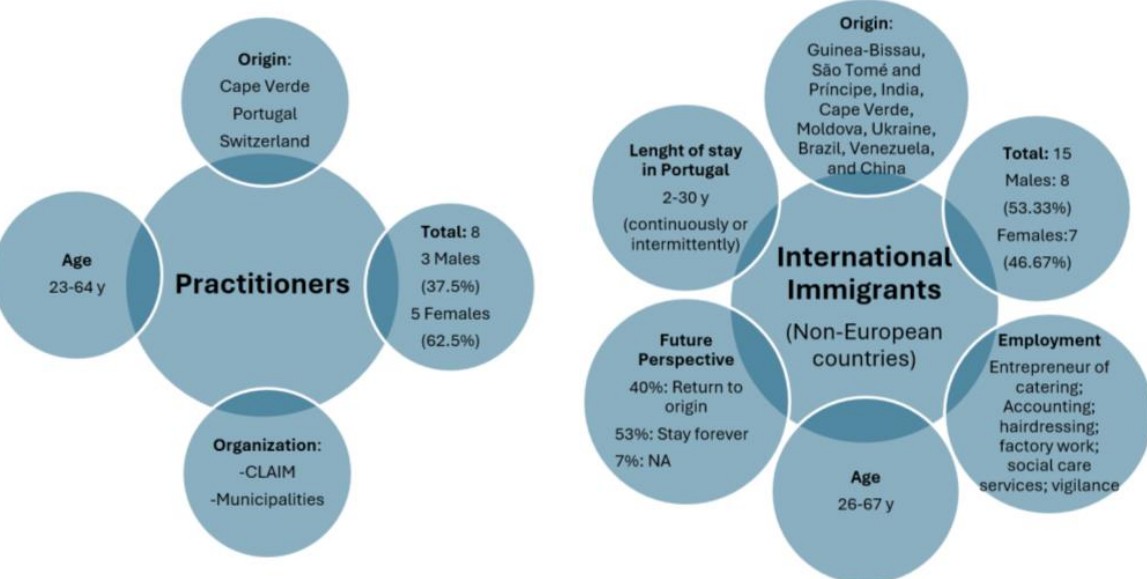

**Figure 1.** Participants of the study. Source: own elaboration.

The second group included international immigrants and non-European Union natives. The participants came from Guinea-Bissau, São Tomé and Príncipe, Cape Verde, Moldova, Ukraine, Brazil, Venezuela, and China, with a maximum variation within the most represented countries in the region [29,30]. Some of the participants had a secondary education level, and others had bachelor's, master's, and even PhD degrees. Their labor was related to accounting, hairdressing, factory work, social services, vigilance, and research. It comprised 8 males (53.33%) and 7 females (46.67%) between 26 and 67 years old. Regarding the length of stay in Portugal, the range is from 4 up to 30 years. Thus, they arrived from 1993 until more recently in 2019. Among the participants, 6 (40%) intend to return to their country of origin, and 8 (53.33%) intend to stay in Portugal. One (6.67%) of the participants was reluctant to answer.

### 2.2. Analysis

Empirical data were transcribed verbatim in Portuguese, except for the data referenced. The task was supported by Express Scribe Pro software version 10.05. Qualitative data from interviews were analyzed, supported by MaxQDA software, and inspired by Braun and Clarke's work on thematic analysis [31].

Coding and categorizing, and then finally, an in-depth interpretation were conducted to obtain immigrants' and practitioners' points of view on the discussed topics [31]. Steps of thematic analysis suggested by Braun and Clarke [31] followed, and lastly, the manuscript was written.

After the analysis and during the writing of this paper, the narratives were translated into English for an international audience.

### 3. Results

#### 3.1. Motivations to Stay Long-Term

Findings on the motives to stay long-term were grouped into four dimensions: (1) Instrumental and material motivations; (2) Emotional and social motivations; (3) Motivations based on quality of life; and (4) Motivations based on political dimensions.

### 3.1.1. Instrumental and Material Motivations

Instrumental and material motivations consist of factors related to basic needs, such as economic aspects, employment, and housing, for example. However, employment is a determinant of remaining in adverse and depopulated regions, meaning that stable and suitable jobs are required for permanency, as well as access to decent and fair housing without racial and cultural discrimination from landlords. Additionally, the quality of both education opportunities and health services for children is seen as an investment for a better family future. Employment is presented by participants as a key factor for an international immigrant to be attracted by Portuguese rural areas and smaller cities. Other factors are not so relevant for them, while this first requisite is not satisfied. Agriculture is the main sector welcoming the immigrant workforce, but in depopulated regions, there are vacancies from all economic sectors beyond agriculture. It is very likely for immigrants to find jobs in the hospitality industry, construction, and cleaning services, as well as qualified jobs such as health professionals and academic researchers. Once these regions present manpower shortages and, therefore, offer jobs for immigrants, immigrants tend to remain in these places. Quotes from participants are consistent and give an understanding that it is unanimous.

> "While there is employment, they [international immigrants] will continue living here in the region [. . .]—In your opinion, what brings migrants to this specific region?—Jobs, job opportunities. They came here because they heard about a job opportunity." [Practitioner 1 CLAIM 1a]

> "Here, there is a lot of agriculture, and because of that, a lot of immigrants." [Immigrant 9, Ukraine]

> "They know that they can easily get a job here, in an orchard, I mean, in the agricultural sector." [Practitioner 3, Municipality 2].

> "However, there is also a shortage of manpower in other sectors beyond agriculture." [Practitioner 4, CLAIM 3].

> "Here, they can also get jobs in hotels, construction industry, house cleaning. . ." [Practitioner 2, CLAIM 2].

> "Employment. . . In the hospital, we have doctors and nurses from Spain and from Brazil. They came to work here." [Practitioner 4, CLAIM 3]

In the Alentejo region, particularly in the agricultural sector, employment tends to be seasonal and, therefore, precarious with several uncertainties. Consequently, when unemployed and with no job opportunities appearing, immigrants tend to leave the region, seeking a job in other municipalities or even in another country. Employment stability could retain immigrants in depopulated regions, according to the interviews.

> "They work seasonally in the production of olives, and then they leave. Then, they come back again for the harvest of red fruits, and so on." [Practitioner 2, Municipality 1]

> "If they cannot find a job here, they leave the region. . . Only a few unemployed immigrants remain here. We know an unemployed family supported by its family and friends and by our organization, but it is becoming unbearable." [Practitioner 1, CLAIM 1a]

Rural regions and smaller cities are more attractive than bigger cities in terms of employment because lower skills are required based on the shortage of manpower. Therefore, once an applicant appears interested in work, he/she is given a job even if his/her experience does not fit the vacancy.

> "Here labor markets are not so demanding, and that is an advantage" [Immigrant 3, Brazil]

"Lisbon is a big city. Employment there can be obtained in civil construction or companies, but speaking Portuguese is required. Thus, for international immigrants to get a job, leaving Lisbon is necessary. Therefore, they come here, and after that, they remain here." [Practitioner 4, CLAIM 3].

"In Lisbon [city], they must speak Portuguese, so the language is a problem. Therefore, they come here to work [namely in agriculture], and then they remain." [Practitioner 3, CLAIM 2]

One of the reasons for the importance of being employed and keeping employed is that it means earning money and sending remittance to the country of origin, namely to pay dues sustained by their family and their projects. For immigrants, it is mandatory.

"They said to me: if I don't send money to my family, they don't eat" [Practitioner 4, CLAIM 3].

"They aim to work extra hours to earn more money and send it to their families in the country of origin" [Practitioner 1, Municipality 1].

Housing availability is another relevant factor in attracting and retaining immigrants in depopulated regions once they gain employment or are seeking it there. However, we found that the availability of houses is not enough, and adequate housing conditions and comfort are demanding, as is the right to access an affordable home and eliminating discrimination from the landlords based only on race and origin before knowing a person. Owing to the scarcity of houses, they tend to be overcrowded, or immigrants are likely to rent accommodation without a housing license, for example, closed restaurants, empty stores, and garage suites. Furthermore, undocumented immigrants are considered illegal, so they do not have the right to housing support from a municipality or national social security benefits.

"- [. . .] I am not sure what could root them here, perhaps housing." [Practitioner 1, Municipality 1].

"There are three or four families living in a three-bedroom house in order to share costs" [Practitioner 1, CLAIM 1].

"They live in a house that isn't properly registered [. . .] it was a restaurant" [Practitioner 1, Municipality 1].

"Shortage of houses and those available are not furnished or are located in the historic center and very deteriorated... However, they rent it. They submitted themselves to those conditions [. . .]. They ask us for beds and even mattresses because they are sleeping on the floor." [Practitioner 3, CLAIM 2]

Housing provided by an employee is seen as an advantage in rural areas since an immigrant may be hosted on the farm itself, where he/she works.

"The vast majority work on farms, and of these, many stay on the farm." [Practitioner 1, municipality 1].

Within the reality described, the participants suggested political and private investment in the region, namely in housing and infrastructure serving immigrant needs to attract and retain them.

"There is an airport, but it doesn't operate. There is a train, but it isn't operating. Thus, while there isn't an improvement in the infrastructure of the region, the municipalities are hampered." [Practitioner 1, Municipality 1].

"[. . .] municipal houses are occupied, therefore if there was a political investment to renovate other houses. . . because there are a lot of unsound houses here, a lot of uninhabited houses [. . .]" [Practitioner 4, CLAIM 3].

Access to quality services and resources, particularly in comparison to their country's resources, is a key factor in wanting to stay long-term. This aspect is even more relevant as

an investment for their children in the present and future, grounded in quality education and health, which they perceived as better, well-equipped, and supportive. Third-national citizens envisage that Europe is the best continent for safety and quality of life, and therefore, the Portuguese rural areas are covered, too. According to non-European Union citizens, these areas are as good as any other European destination.

> "It is possible for you to have your own family here and to give a good education to your children. . . There are hospitals, a good hospital. And the local health center is good as well. And schools. . ., here there are good schools and even good universities." [Immigrant 5, Brazil].

> "She said to us: I want to give my children a European education. I want my children to access health services. I want them to have educational opportunities that I could never give in my home country" [Practitioner 4, CLAIM 3].

3.1.2. Emotional and Social Motivations

The so-called emotional and social motivations dimension is grounded in factors related to socio-emotional satisfaction. Family presence and wellbeing can achieve this satisfaction, as can engaging in migrants' networks, being really integrated with locals, and establishing close relationships with social services.

The second component consists of an emotional dimension that complements the instrumental aspects presented before for an immigrant wanting to stay in these regions. With stable employment and a house, the next step, which is linked to relationships with their own family and others, may be arranged. Thus, emotional and social factors of motivation arise in both personal and family wellbeing. Being well and having their own family around to provide emotional support is central. Participants said that the family represents an important value for immigrants. Therefore, family reunification is also an instrument to integrate and retain immigrants in depopulated regions. Immigrants who have already brought their children to Portugal are more likely to remain where they are to provide more comfort and family stability that perhaps was not possible in the past.

> "They want to establish here, bring their own family, bring their wives and children who remain in the country of origin. Sometimes, they bring their family in dribs and drabs; now comes the wife, then one son. Men come first to get a job, and later come family." [Practitioner 3, CLAIM 2].

In addition, the social integration level of an immigrant in the host community is relevant to their desire to remain. Here, integration is discussed based on strong ties to neighbors and the local community and a sense of belonging, as well as the relationship established with practitioners and social workers from local services and organizations that work with migration issues. Professional relationships are built, and the quality of practices is demanded for immigrant satisfaction.

> "Immigrants tend to create a close relationship with the local community, so if an opportunity appears, they are informed and move there. That's called integration!" [Practitioner 4, Municipality 2].

> "Once arrived, newcomers get to know the municipality and other services, where to ask for help and advice. The local capacity to welcome new residents makes them want to stay long-term. And then, relationships with locals. . . they build their relationships as all of us, and. . . establish a connection to local people and services." [Practitioner 4, CLAIM 3].

> "We, practitioners, work very close to them [immigrants] and become a reference for them, someone who they can trust. From then on, they always come back here because they know if the issue is not up to us, we refer them to the suitable service!" [Practitioner 3, municipality 2].

> "Immigrants once here [in Alentejo] want to become legal. Therefore, if there is an organization to support them become regularized, well. . . [even better]. If

support is provided in this region, they [immigrants] will want to stay. And, of course, if they get a job!" [Immigrant 1, Guinea-Bissau].

Migrants' informal networks allow them mutual aid and peer support, namely for those who lack their own household. There is a tendency for migrants to share their experiences with their compatriots, which is stimulating. Then, they bring those compatriots to the same destination, and that supports sustainability since it means new inhabitants for rural regions.

"The household pulls its friends to the region. And if they get a job, they stay long-term." [Immigrant 3, Brazil].

"A person [immigrant] who comes here is following another one [immigrant]." [Immigrant 6, Brazil].

### 3.1.3. Motivations Based on Quality of Life

Motivations based on the quality-of-life dimension are founded on reasons such as lower rents compared to other destinations, a quiet and secure environment, and the guarantee of a better future for their children.

Thus, this third component of reasons for non-EU-citizen immigrants reaching and staying in these regions starts with the perception of quality of life, peace, and a sense of security. These constitute important dimensions that they lack in their country of origin and perhaps in Portuguese big cities. Security and tranquility are seen as an investment and a priority for their children. Children are a great motivator to tackle migration challenges. Immigrants accept having to make sacrifices for them.

"They stay long-term because of the quality of life" [Practitioner 2, CLAIM 2].

"Here, I walk on the street fearless [. . .] secure at any hour, day or night." [Immigrant 10, Cape Verde].

"They [immigrants] told me that after his son-in-law had been murdered on the street, it was impossible to keep living in Brazil." [Practitioner 4, CLAIM 3].

Second, the cost of living is cheaper in rural areas than in big cities. Participants highlighted lower rental costs, which are higher monthly expenditures.

"I was invited by my sister to leave Alentejo and move to Lisbon. Then, I reflected on the pros and cons because of the rental costs..." [Immigrant 7, São Tomé e Príncipe].

"Here is a very calm place, and life is pretty cheap" [Immigrant 13, Guinea-Bissau].

Reviewing their trajectories, immigrants consider that closer relationships and integration, easier networks and support from the community, and economic stability, perhaps by remote work, are advantages of small places that provide a standard of living.

"So, we become friends [immigrants and locals] within a calm way of life." [Immigrant 11, Brazil].

"Here is a calm place with generous local inhabitants. . . The welcome that we get here. . . International residents wouldn't receive it in many places." [Immigrant 13, Guinea-Bissau].

### 3.1.4. Motivations Based on Political Dimension

Finally, motivations based on political dimensions are grounded in factors associated with the simplified procedure to legally live in Portugal.

The fourth component found in participant discourses raises the open-door migration policy framed to combat workforce shortages in Portugal, considering that demographic issues are present in the whole country.

An open door for international immigrants from third countries is the most stimulating factor in choosing Portugal as a destination. Portugal is known for its simplicity in the

regularization procedure to become a legal resident when compared to other Western countries. Obtaining nationality, and consequently European citizenship, is also a simplified procedure in Portugal. This allows a third-national person to move and live in another European country and earn higher wages.

> "The regularization is faster. And even if they [immigrants] must wait for three or four years, they know that for sure they will get the document. Third-country nationals may enter Portugal lacking proof of legal entrance, since the law amendment." [Practitioner 3, Municipality 2].

> "The regularization in Portugal is very... let's say, more simple, less complex than in Italy, Spain or Germany." [Practitioner 3, Municipality 2].

There are several criteria enabling authorization; participants stated the option of becoming legal through employment or their descendants.

> "I always tell an immigrant: if you want to live here, the first thing to do is have babies. They may request authorization for residence in the interest of the child once having a baby born in Portugal." [Practitioner 1, CLAIM 1].

In smaller cities, services tend to deal with a lower number of service users, and perhaps responses are more prompt.

> "Here, the border service used to address half a dozen immigrants per week, and now it is half a dozen per day. [...] 90%, almost 100% of the service users are from abroad." [Practitioner 2, CLAIM 2].

In summary, participants presented their perspectives on motivations that influence immigrants to stay in Portuguese-depopulated regions and smaller cities, such as in the Alentejo region. The practitioners and immigrants interviewed discussed factors that are attracting immigrants to be there and what could retain them if political investments were made. Therefore, Figure 2 includes participant suggestions for policy and practice, as well as for host communities.

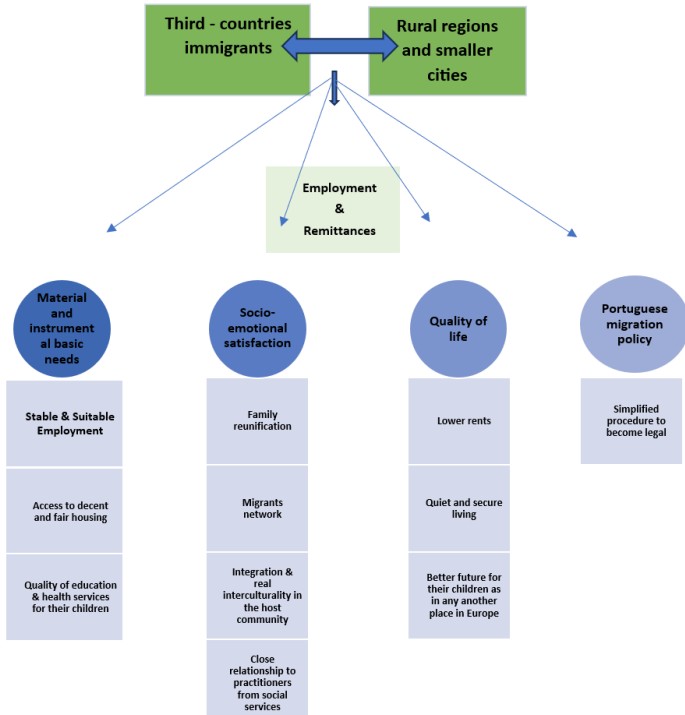

**Figure 2.** Diagram of factors influencing immigrants' motivation to remain in rural areas. Source: own elaboration.

Our thesis counters the common perception that immigrants will stay in rural areas for employment and that if employment is ensured, they will remain for money and remittance as long as it is warranted. Furthermore, this study found that immigrants' basic needs are more than employment. They desire stable and suitable jobs, access to decent and fair housing, as well as quality education and health services for their children. These, in addition to socio-emotional satisfaction, include family reunification, migrant networks, integration and real intercultural living in the host community, a close relationship to practitioners from social services, and finally, quality of life, such as lower rents, a quiet and secure environment, and a better future for their children, as in any other part of Europe. Additionally, the simplified policy to become legal encourages an immigrant to remain, which is a constructive ongoing process.

Policy and practice must be planned strategically to address ecosystem needs and to be more worthwhile when an immigrant balances their current situation and their perception of what alternative scenarios are present in medium-sized and bigger cities.

## 4. Discussion

This study explores the motivations shaping immigrants' intentions to stay in rural regions and smaller cities, particularly in Alentejo, which is characterized by depopulation issues. Practitioners and immigrants described a set of factors that they believe are influencing and could influence third-country-national immigrants to remain in the region.

The study design and results are innovative, as scholars are paying little attention to this specific topic; therefore, results on the motivations for living in other places are discussed in the context of literature.

As presented, the results state that staying permanently in rural regions and small cities, such as Alentejo, may be a constructive process and not an objective and previous aim. International immigrants tend to plan to move short-term to a region, attracted by several factors, particularly related to employment. Then, if a couple of needs (individual and/or familiar) seem to be met, the idea of a long-term stay in the region may develop. That is aligned with Flynn and Kay's [24] discussion of rural Scotland, saying that long-term becomes the foreseeable future plan. Therefore, rural policymakers and governments are called to strategically stimulate international immigrants' motivation to develop the idea of staying long-term in a rural municipality.

Therefore, we argue that immigrants' needs, identified in the results, must be addressed by local and regional policy, as according to Flynn and Kay, immigrants reflect and balance their experiences and aspirations for the past, present, and future [24] and thus, the foreseeable risks and uncertainties that Valdez and colleagues note [25]. We conclude that in rural regions, immigrants remain if assurance and certainties are higher than in other destinations.

Employment and economic migrants are unanimously referred to in findings, and in the literature related to the migration phenomenon, e.g., [10,23], but our study found that employment by itself is not enough for a newcomer to live in an adverse place, such as a rural region. Therefore, to remain, first employment should be stable and suitable, or they will seek to leave as soon as a better opportunity appears elsewhere. Then, other basic needs must be addressed, such as access to suitable and rentable housing. Access to quality education and health care services for their children is essential for a third-country national living in rural Alentejo. It could be said that it is relevant for any immigrant with children since it is well known that a reason for citizens to leave a country is the prospect of a better life for their children, namely European government-sponsored services and resources [7,25].

Family-focused motivations to leave and to remain are present in all discourses as a relevant factor to a decision, as well as complementary, informal networks [7,10] that may attract foreigners and then facilitate their social integration in a new place. Gauci [10] highlights this in his study of several countries, such as Belgium, Bulgaria, Germany, Italy, and Sweden, regarding key findings for the integration of migrants in medium-sized and

small cities and rural areas in Europe. International migrants benefit from having access to rural regions and smaller cities' closer networks and increased interactions with the local population [10].

However, emotional satisfaction determinants for third-country immigrants wanting to stay in an adverse place, as well as the relevance of building a close professional relationship with social workers who work in organizations for migration, are motivational factors not sufficiently explored in the literature. Increasing psychological aspects are becoming relevant. Rural areas and smaller cities may provide a stronger tie between newcomers and host communities. These regions may be prepared to reduce integration efforts with lower rents, greater community support, less traffic, and thus less time spent, as highlighted by Gauci [10].

Four components were constructed in this study, grounded in participant interviews after a thematic analysis: instrumental and material motivations, emotional and social motivations, motivations based on quality of life, and motivations based on political dimensions. Valdez et al. presented psychological or personal, social, and political factors to remain in an adverse destination in an anti-immigration state in the USA [25]. They found personal motivations, such as perseverance, resignation, fatalism, and religious faith, that did not emerge in our study. However, family and community factors, such as dreams and hopes for their children, access to community resources, and social ties, are factors aligned with our results. Valdez et al. also discussed the sense of belonging and feeling connected to the community, similar to Chavez [26] in his study about undocumented Mexicans and Central Americans in the USA. By contrast, our participant narratives remotely touch on those concepts. Perhaps a reason for the disparity between the findings of these studies could be the length of the permanence in the host community, language similarities, cultural differences between newcomers and locals, and even local, regional, and national policies.

Gauci [10] states that "in many cases, the number of refugees moving to small cities is determined by national policy through dispersal policies, for instance" (p. 30). Thus, based on that policy view, we recommend a similar strategy for all international immigrants within a regional and local policy to develop rural areas grounded in a strategic management of migratory flow. A pilot could provide insights for a second phase at a national and European level.

Rural areas need to consider it worthwhile to invest in integration to retain immigrants. Both immigrants and host communities must address integration needs on the basis that immigrants are there to stay; therefore, longer-term engagement and integration are worth the investment of resources, time, and effort. [10]. Accessing services, such as healthcare, are relevant. Therefore, these services need to be accordingly equipped to deal with the continuous increase of immigrants [8,12,21,22].

Based on its results, the present study recommends designing local policies within a strategic plan that includes actions intentionally addressing international immigrant issues to stimulate their retention, as well as their families, in depopulated regions. Specifically:

(1) Awareness of the labor market/ employers for (a) the rights of immigrant employees, as permanent contracts to keep them for long-term; (b) valuing cultural diversity; and (c) recognizing previous education and experience.

(2) Encouraging private investment for the renovation and offer of houses; mediating rental contracts; avoiding contact between landlords and immigrants, and therefore discrimination; providing legal support in return for contentiousness.

(3) Reinforcing non-governmental migrants' rights organizations to guarantee information, counseling, and referrals for everyone whenever needed.

(4) Promoting activities for intercultural dialogue and a sense of community in host societies.

(5) Stimulating family reunification for long-term permanence and increasing population.

(6) Providing more public transport and accessibility. For social inclusion, more infrastructures, and more services, local policymakers should mobilize the European

Union budget by applying for the European Social Fund+, the European Regional Development Fund, Erasmus+, and other foundations (European Commission, 2020).

The deepened knowledge presented in this study may inspire other European (and non-European) countries that are facing similar challenges and contribute to the Sustainable Development Goal (SDG) of the 2030 UN Development Agenda that pays explicit attention to rural issues.

## 5. Conclusions

Europe is facing an internal demographic challenge that may have serious consequences for the sustainability of its social model because of aging. For rural areas, the challenge is even greater. Migrants who want to come to Europe may represent a resource for these areas once they become attractive, welcoming, and sustainable. This study investigates how Low-Density Territories can retain third-country citizens.

The main lesson is that since both parties profit, two-way investment is strategic. Thus, policy and practice are needed on the one hand to identify and satisfy new arrivals' needs and, on the other hand, to provide host-society awareness; thus, both parties must be involved as active citizens. Early and effective integration is mandatory for an inclusive community. The findings of this study present reasons for immigrants wanting to stay in a region that translates to satisfaction.

Results show four components of motivation: instrumental and material motivations, emotional and social motivations, motivations based on quality of life, and motivations based on political dimensions. Moreover, that motivation for longer permanency in a rural region is not a pre-migration wish. By contrast, it is a likely progression composed of a set of factors that produce the desire for long-term permanence.

Implications target: (1) Policymakers at local and regional levels (as well as central government) should plan a set of strategic actions to satisfy immigrants' needs and retain them and their families in depopulated regions for sustainable development. (2) For practitioners who are on the front line of social policies, we recommend building a very close support relationship grounded in an eco-systemic framework and fundamental rights approach. (3) Host communities must be aware of (and be made aware of) migration challenges and be available to foster and provide intercultural environments for the richness and development of their territories. Multidimensional interventions must address complex problems, and therefore, planned actions should be evidence-based and supported by academia.

### Challenges and Future Perspectives

Some limitations of this study are presented, namely in the representativeness, based on the small number of participants, since it is a qualitative approach. However, as indicated before, theoretical saturation was assured. Future research within a mixed-methods approach could provide a cross-country view of the topic in rural Europe, considering that the issues presented are also present in the entirety of Europe.

Therefore, more in-depth research is needed, namely on the impact of immigrants in Low-Density Territories in terms of demographic, social, economic, cultural, and religious factors that contribute to facing their challenges and sustaining rural regions. Therefore, more data ought to be collected and framed into future research to address the evolution of immigration and its local impact, aiming to inform local, national, and European policy. Quality-based policy and action for welcoming, integrating, and retaining immigrants are crucial to building an inclusive society for social cohesion in Europe [13,32] and achieving Sustainable Development Goals.

**Funding:** This research is part of the project called Ir Além—A Inclusão Social de NPT e o Desenvolvimento de Territórios de Baixa Densidade, co-funded by the Asylum Migration and Integration Fund, operation number PT/2020/FAMI/535". The project was awarded by the Centro de Estudos Ibéricos with the Research, Innovation, and Territories Award 2023. This work was also supported

by national funds through the Fundação para a Ciência e a Tecnologia, I.P. (Portuguese Foundation for Science and Technology) by the project UIDB/05064/2020 (VALORIZA—Research Centre for Endogenous Resource Valorization).

**Institutional Review Board Statement:** Not applicable.

**Informed Consent Statement:** Informed consent was obtained from all subjects involved in the study.

**Data Availability Statement:** Data are available on request.

**Acknowledgments:** We acknowledge the financial support given by the Asylum Migration and Integration Fund (AMIF)—European Commission; the administrative and technical support from the research and innovation office and from the whole Polytechnique University of Portalegre; and finally, the collaboration of volunteers, research fellows, and partners. Our main words are to the immigrants and practitioners who were available to participate in the present research.

**Conflicts of Interest:** The author declares no conflicts of interest.

## Notes

[1]  In this paper, the terms migrant/immigrant/migration are used to refer to third-country nationals (TCN), i.e., any person who is not a citizen of the European Union within the meaning of Art. 20(1) of The Treaty on the Functioning of the European Union—Consolidated Version of. Official Journal of the European Union. 2012. https://eur-lex.europa.eu/legal-content/EN/TXT/PDF/?uri=CELEX:12012E/TXT&from=EN (accessed on 30 January 2024). Therefore, this may include a person with or without residence permission.

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
