# Peer review of "Retaining Non-EU Immigrants in Rural Areas to Sustain Depopulated Regions: Motives to Remain"

_societies, doi:10.3390/soc14020025_

Round 1

Reviewer 1 Report

Comments and Suggestions for Authors

 The article is interesting and relevant to today's rural issues, not only in Portugal but also in the entire Europe. Although the article includes all the necessary sections required for academic papers, the reviewer suggests some changes that could enhance this work.

1.       Abstract is a bit chaotic, introducing too much of facts.

2.       Keywords are all phrases; to reviewers’ opinion the “keyword” “Keeping people in Low density territories” is especially too long. And why some words are capitalized?

3. The Introduction section appears more like a brief literature review. It would be beneficial for the authors to clearly identify the objective of the paper, perhaps by explicitly naming it as the 'objective.' Additionally, there is a need to present the novelty more distinctly. The research questions are reiterated three times: in the abstract, introduction, and Material and Methods sections.

 Moreover, the reviewer recommends transferring the last paragraph of the Introduction section (lines 159-162) to either the Acknowledgments or Funding statements or the endnotes/footnotes. To enhance clarity, it is suggested to create a separate brief introduction section at the beginning and a distinct literature review section. Alternatively, the literature review information can be incorporated as a clearly indicated subsection within the introduction.

 The Introduction section contains some inconsistencies that could be clarified for better coherence. Initially, the text emphasizes 'great depopulation' in the region, but shortly thereafter, it mentions an increase in 'considerable numbers.' To enhance clarity and coherence, it is recommended to provide more explicit and consistent statements regarding the demographic trends discussed in this section.

 Example: “In the region there was a decrease of inhabitants in 2021, around 7.0% of 704,533 people” (line 65) and “[…] immigrants are largely living in urban areas, rural areas have increasingly begun to attract immigrants in considerable numbers [9], as in the Alentejo for example. In 2021 there were 23,737 (19.8%) newcomers [18]. In total, there was a total of 52,316 immigrants” (lines 85-87)

4. In the Material and Methods section, there is unnecessary repetition of the number of participants. It would be more insightful for the readers to understand the rationale behind the diverse selection of participants and why specific nationalities were chosen. Providing this context would enhance the depth of the research and contribute to a better understanding of the study's design.

5. In the Results section, the introductory paragraph (lines 215-219) repeats information already covered in the Introduction and Methods sections, making it redundant. Therefore, it is unnecessary. Additionally, the content in lines 220-223 would be better placed under the 3.1. subtitle.

 As mentioned in the Introduction section, this research is part of a larger international project. To enrich the discussion of the results, it would be worthwhile to include sentences with examples from other countries and compare them with the Portuguese situation. This comparative analysis could be incorporated either in the Results section or in the Discussion/Conclusion.

6. The Discussion section appears weak as it seems to repeat statements from the Introduction and intertwines with the Results text. To improve coherence and provide a more meaningful discussion, it might be beneficial to integrate the sentences from the Discussion section into the Results section, specifically in the context of comparing the findings with other studies. This approach would enhance the flow of the paper and offer a more seamless transition between presenting results and discussing their implications in relation to existing literature.

If it is deemed that the answer to the question 'What contributions can practitioners and immigrants give for local policy and practice?'—as posed in the Introduction section—should be found in the last paragraphs of the Discussion section, it implies the need to expand and structure this answer more comprehensively. This would make the answer to the question more visible.

Comments on the Quality of English Language

 It is recommended that the English language be revised by a native speaker. This would improve the readability and comprehension of the text. The text throughout the entire article is not written in an appropriate English style.

In addition, the reviewer suggests conducting a thorough check of the translated citations, as they exhibit grammar and style issues, and some essential words may be missing. .

 For instance: “it is a house that even has (???) housing license […] it was a restaurant” (line 296) or “immigrants once here, want to be illegal (???), so, if an organization exists to support them, well…[even better]. They will want to stay, and of course if they get a job” (lines 360-361). the reviewer notes that some words are missing or translated incorrectly in the citations, resulting in a lack of alignment with the preceding research text. 

Author Response

Dear Reviewer,

Reviewer 2 Report

Comments and Suggestions for Authors

The manuscript societies-2745575 is devoted to the actual scientific problem, namely study of the motivations shaping immigrants’ intentions to stay in rural regions and smaller cities. The reviewed article is interesting for scholars and theme of the article meets the scope of the journal. Work is performed at sufficient scientific level and has good quality. The manuscript may be considered for publication after major revision in Societies. Prior publication of this manuscript following points needs to be addressed:

  • It would be good to broaden the Discussion in the context of comparing the obtained results with the data of similar studies, especially, if possible, with other countries. The above discussion is of a local nature, and it does not correspond to the level of an international journal.
  • The topic discussed by the authors is quite debatable and can have many directions for development. I propose to add separate section "Challenges and Future Perspective".
  • Many quantitative data are provided in the introduction and Materials and Methods. This raises the scientific level of the work, so for a better understanding of the material, it is worth presenting such data in the form of diagrams.
  • Minor editing of English language required

My decision is major revision

Comments on the Quality of English Language
  • Minor editing of English language required

Author Response

Dear Reviewer,

Round 2

Reviewer 1 Report

Comments and Suggestions for Authors

The author considered the feedback from the reviewer, and the article could be accepted for publication

Reviewer 2 Report

Comments and Suggestions for Authors

The authors took into account the comments. A revised manuscript may be accepted.